# Fresh Tomato (*Lycopersicon esculentum* Mill.) in the Diet Improves the Features of the Metabolic Syndrome: A Randomized Study in Postmenopausal Women

**DOI:** 10.3390/biology13080588

**Published:** 2024-08-03

**Authors:** Chein-Yin Chen, Yi-Wen Chien

**Affiliations:** 1School of Nutrition and Health Sciences, Taipei Medical University, Taipei 11031, Taiwan; cyc.charlotte@gmail.com; 2Graduate Institute of Metabolism and Obesity Sciences, Taipei Medical University, Taipei 11031, Taiwan; 3Nutrition Research Center, Taipei Medical University Hospital, Taipei 11031, Taiwan; 4Research Center of Geriatric Nutrition, College of Nutrition, Taipei Medical University, Taipei 11031, Taiwan; 5TMU Research Center for Digestive Medicine, Taipei Medical University, Taipei 11031, Taiwan

**Keywords:** postmenopausal women, tomato, metabolic syndrome, antioxidant biomarkers

## Abstract

**Simple Summary:**

To investigate the impact of tomato consumption on reducing metabolic syndrome risk factors in overweight postmenopausal women, we conducted a randomized controlled trial with an 8-week open-label dietary intervention. Overweight postmenopausal women aged 45–70 were randomly divided into two groups: a control diet and a tomato diet. The tomato diet group showed significantly lower body fat mass, body fat percentage, waist circumference, and hip circumference compared with the control group. They also had significantly lower serum total cholesterol, triglycerides, systolic blood pressure, and blood sugar levels, as well as higher high-density lipoprotein cholesterol. Additionally, antioxidant biomarkers such as FRAP, beta-carotenoids, and lycopene were significantly higher in the tomato diet group. These findings suggest that fresh tomato consumption can enhance antioxidant biomarkers and reduce metabolic syndrome risks in postmenopausal women.

**Abstract:**

(1) Background: According to the 2005~2008 Nutrition and Health Survey in Taiwan (NAHSIT), more than half of Taiwanese women (57.3%) had metabolic syndrome during menopause. Metabolic syndrome is a set of risk factors for cardiovascular disease (CVD) that increase the risk of cardiovascular disease, diabetes, and mortality. Epidemiological studies suggest that the consumption of tomato-based foods might reduce the risk factors for CVD. The aim of this study is to examine the effects of tomato consumption on lowering the metabolic syndrome risk factors among overweight postmenopausal women. (2) Methods: We conducted a randomized controlled trial using 8-week open-label dietary intervention. Overweight postmenopausal women aged 45–70 years old were recruited from Taipei Medical University in October 2013. They were randomly assigned into two groups (a control diet vs. a tomato diet). Blood samples were collected at the baseline and at the 4th and 8th weeks. The lipid profile, blood sugar, and antioxidant biomarkers, i.e., the ferric-reducing ability of plasma (FRAP) and serum carotenoids, were analyzed. Blood pressure, body weight, and body fat were also measured every week. (3) Results: After the 8-week dietary intervention, body weight, body mass index, waist circumference, and hip circumference were significantly lower in both groups (*p* < 0.05). Body fat mass, body fat percentage, waist circumference, and hip circumference were significantly lower in the tomato diet group than in the control diet group. The tomato diet group had significantly lower serum total cholesterol, triglyceride, systolic blood pressure and blood sugar, and higher high-density lipoprotein cholesterol than the control diet group. The antioxidant biomarkers, FRAP, beta-carotenoids, and lycopene were significantly higher in the tomato diet group than in the control diet group. (4) Conclusions: Fresh tomato consumption can increase antioxidant biomarkers to reduce risks of metabolic syndrome in postmenopausal women.

## 1. Introduction

According to the World Health Organization (WHO), in 2019, heart and cerebrovascular disease ranged as the second and fourth causes in the world’s top 10 leading causes of death. In Taiwan, the top 10 causes of death related to cardiovascular disease (CVD) are at 23.0%. Metabolic syndrome (MetS) refers to a series of cardiovascular risk factors, including hypertension, hyperlipidemia, high blood sugar, and abdominal obesity, that could increase the risk of cardiovascular disease and diabetes [1].

Epidemiological studies suggest that the consumption of tomato-based foods might reduce CVD risks [2,3,4,5]. The consumption of seven servings/week of tomato-based products could reduce by 30% the relative risk of CVD [6]. Postmenopausal women likely have abdominal fat and visceral fat accumulation, as well as increased CVD risk factors [6,7,8]. Tomatoes are rich in lycopene, carotenoids, and a variety of phytochemicals and are extensively consumed worldwide [9]. Serum carotenoids and lycopene are negatively correlated with indicators of inflammation and vascular endothelial dysfunction [10]. A number of studies focused on foreign tomato species or tomato-based processed products [11]. A human clinical study showed there is a clear connection between tomato supplementation and positive effects on human biochemical parameters (such as blood glucose, HbA1c, harmful lipid profile, inflammatory markers, and free radicals), which is likely to reduce obesity, diabetes, and cardiovascular events [12].

So far, the beneficial effects of fresh tomato (*Lycopersicon esculentum* Mill., a native Taiwan variety) consumption on metabolic syndrome in overweight women after menopause have remained unclear. Black persimmon (*Lycopersicon esculentum* Mill.) is the most traditional tomato variety in Taiwan. The pulp of black persimmon tomatoes is relatively hard and has high acidity, so it is very suitable for stir-fries, soup bases, or tomato scrambled eggs. Its sour taste can add a special flavor to dishes. In Kaohsiung and Tainan in southern Taiwan, black persimmons (*Lycopersicon esculentum* Mill.) are also cut into plates and mixed with minced ginger and soy sauce to become a famous snack “tomato cut plate”. Therefore, this study aimed to evaluate the effects of fresh tomato consumption on the risk factors of metabolic syndrome among overweight postmenopausal women.

## 2. Materials and Methods

### 2.1. Study Design and Participants

We conducted a randomized controlled trial for 8 weeks on 60 overweight postmenopausal women aged 45~70 years, with a body mass index (BMI) > 24 kg/m^2^. Participants were recruited from the community of Xinyi District, Taipei City, Taiwan who visited Taipei Medical University Hospital in October 2013. Study participants were assigned to two groups with 30 people in each group. Subjects were asked to consume a control diet or a tomato diet. This study was approved by the Taipei Medical University Research Ethics Committee (no. 201307033).

Individuals with metabolic syndrome or moderate hypercholesterolemia were eligible. We excluded women diagnosed with CVD, diabetes, asthma, a thyroid condition, eating disorders, high habitual intake of tomatoes and tomato-based products, or taking regular medication or supplements known to affect any dependent variables.

### 2.2. Experimental Diets

We conducted a dietary intervention on two groups, a control diet and a diet with two servings of tomatoes per day (corresponding to an estimated minimum of 11 mg of lycopene/day). During the dietary experiment, both groups consumed 1500 kcal, consisting of 18% protein, 28% lipids, and 54% carbohydrates. Two groups had a recommended daily intake of five servings of vegetables and two servings of fruit. Two servings of vegetables in the tomato group were replaced with tomatoes (about 200 g/day), and raw tomatoes were eaten with breakfast and lunch. Tomatoes were provided weekly from the laboratory. The control group used sprouts as an intervention. The control diet group consumed a regular diet, and the tomatoes and tomato-made product intakes were restricted. Participants in the control group were not allowed to consume any of the forbidden foods, such as pasta, canned tomatoes, cooked tomatoes (e.g., fried or grilled), tomato paste, tomato puree, pizza, salsa, and spaghetti. They were allowed to consume up to 1 portion of tomato soup, tomato juice, or tomato sauce, 4 raw tomatoes, or 24 cherry tomatoes per week. Dietary intake was assessed using 7-day food diaries before and during the run-in period, as well as during the intervention. Dietary diaries were analyzed using Nutrition professional software (verse. 1). The control group contained no vitamin A or lycopene.

### 2.3. Data Collection

Twelve-hour fasting blood samples were collected at baseline and the 4th and 8th weeks and were analyzed in Taipei Medical University Hospital for follicle-stimulating hormone (FSH), estradiol (E2), blood lipid profile, blood sugar, and insulin. The blood lipid profile consisted of total cholesterol (TC), triglyceride (TG), high-density lipoprotein cholesterol (HDL-C), and low-density lipoprotein cholesterol (LDL-C). Homeostatic model assessment for insulin resistance (HOMA-IR) and the quantitative insulin sensitivity check index (QICKI) were used to estimate insulin resistance and sensitivity. Blood pressure, body weight, and body fat were also measured every week. Blood pressure was measured using a Beurer BM16. Body composition was measured using InBody v3.0. Plasma vitamin A, vitamin C, lycopene, vitamin E, and beta-carotene were measured by reverse-phase high-performance liquid chromatography (HPLC) on a Hitachi D-2000. The antioxidant ability was analyzed by the ferric-reducing ability of plasma (FRAP) using a commercial kit (Abcam, Cambridge, UK, catalog no.ab234626) with an enzymatic colorimetric test. The inflammatory biomarker, interleukin (IL)-6, was analyzed by a sandwich enzyme-linked immunosorbent assay (ELISA). Using a Biolegend (San Diego, CA, USA) ELISA kit with precoated plates, it was determined that the stronger the inflammation in the plasma, the higher the concentration of IL-6, the darker the yellow color at 570 nm, and the higher the OD value. 

Questionnaires were used to assess their smoking, drinking, and exercise habits. Smoking and drinking included the frequency, amount of food consumed each time, and type. Exercise habits recorded the exercise items and time. Participants exercised regularly before and after the experiment.

### 2.4. Statistical Analysis

The Shapiro–Wilk test was conducted for a normal distribution test. In the one-way analysis of variance (ANOVA), differences in the comparison group at 0, 4, and 8 weeks were assessed by Duncan’s new multiple-range post hoc test. Differences between the control diet group and the tomato diet group at each time point were assessed with independent sample *t*-tests. A paired *t*-test was used to compare differences in each group at 0, 4, and 8 weeks. SPSS v19.0 (SPSS, Chicago, IL, USA) was used for the analysis with *p* < 0.05 indicating statistical significance.

## 3. Results

### 3.1. The Nutrients of Control and Tomato Diet

Table 1 shows the nutrients of the control and tomato diets. There were no significant differences in crude fiber or dietary fiber between the control diet group and the tomato diet group. On a 100 g basis, the tomato group contained 84.2 RE vitamin A and 5.5 mg lycopene, whereas the control group contained no vitamin A or lycopene.

### 3.2. Baseline Characteristics and Dietary Intake of Subjects

The baseline characteristics of the study participants are shown in Table 2. No significant differences in the baseline characteristics of the study participants were seen. All participants (53 postmenopausal women with metabolic syndrome; 7 dropped out in the control group) completed the entire study. Calculated nutrient content and food group servings of 7 d diet food diaries according to the subjects’ report are shown in Table 3. There was no significant difference in calories during week 0, at 1639.0 ± 61.7 and 1607.3 ± 72.7 kcal, between the two groups (*p* = 0.173). Calorie intake in week 4 after the intervention in both groups significantly decreased to 1606.9 ± 68.7 and 1557.3 ± 66.4 kcal (control group *p* = 0.024 and tomato group *p* < 0.001), while at 8 weeks after the intervention, both groups had significantly decreased to 1571.0 ± 64.9 and 1522.2 ± 60.5 kcal (control group *p* < 0.001 and tomato group *p* < 0.001). After 8 weeks of dietary intervention, both groups had lower energy intakes. There was no significant difference in the control group after the intervention in terms of protein, fat, or carbohydrate intake. However, fat and carbohydrate intake at week 8 was significantly reduced compared with week 0 in the tomato group. There were no significant differences between the two groups in terms of crude fiber intake in the control group and the tomato group at weeks 0, 4, and 8 (*p* = 0.997, *p* = 0.332, *p* = 0.420, respectively).

All values are mean ± SD. Differences between groups were assessed using an independent *t*-test (*p* < 0.05 indicates statistically significant). BMI is body mass index; WC is waist circumference; HC is hip circumference; WHR is waist–hip ratio; SBP is systolic blood pressure; DBP is diastolic blood pressure; FSH is follicle-stimulating hormone; and E2 is estradiol.

### 3.3. Glycemic-Control-Related Markers

Table 4 indicates no significant difference between the two groups in fasting glucose, with the control group at 101.7 ± 33.1 mg/dL and the tomato group at 103.1 ± 13.2 mg/dL (*p* = 0.645). At week 8, there was no significant difference between the two groups at 103.0 ± 24.0 and 98.5 ± 13.3 mg/dL (*p* = 0.975), but the tomato group had significantly decreased compared with week 0 (*p* = 0.0 22). As for the insulin concentration, the HOMA-IR and QUICKI did not change with the intervention between the two groups at weeks 0, 4, and 8.

### 3.4. Oxidative-Stress-Related Markers

As shown in Figure 1, plasma FRAP concentrations were not significantly different between the two groups at baseline (*p* = 0.704), but they were higher in the tomato group than in the control group at week 8 (*p* < 0.001). At week 8, FRAP was significantly decreased from 627.61 ± 137.52 μM/L to 485.6 ± 192.4 μM/L in the control group (*p* = 0.012) and increased from 611.81 ± 147.98 μM/L to 1066.4 ± 609.6 μM/L in the tomato group (*p* = 0.002). There was no significant difference in serum vitamin A concentration between the two groups at week 0 (*p* = 0.740) and week 8 (*p* = 0.715). Serum β-carotenoid concentration was not significantly different between the two groups at week 0 (*p* = 0.112) but differed at week 8 (*p* = 0.014). At baseline, serum lycopene concentration was 28.63 ± 0.77 and 31.48 ± 1.13 μg/mL in the control group and the tomato group, and there was no significant difference between the two groups. However, serum lycopene concentration was significantly higher in the tomato group than in the control one at week 8 (*p* = 0.003). After 8 weeks, plasma lycopene concentrations in the control group were 29.85 ± 0.13 μg/mL (*p* = 0.265), with no significant change, while that of the tomato group had significantly increased by 53%, to 48.25 ± 6.15 μg/mL (*p* = 0.002).

### 3.5. Anthropometric Measures, Blood Pressure, and Lipid Profile

As shown in Table 5, after 8 weeks of the dietary intervention, body weight, BMI, fat mass, fat mass percentage, waist circumference, and hip circumference in both groups had decreased. Moreover, the body fat mass, fat mass percentage, waist circumference, and hip circumference of the tomato group had significantly decreased compared with the control group. In the control group, the starting point of the body fat mass was 24.7 ± 7.3 kg, then 24.5 ± 8.0 kg at week 4, and 24.3 ± 7.5 kg at week 8. In the tomato group, corresponding values of fat mass were 22.7 ± 3.1, 22.3 ± 2.8, and 22.0 ± 2.7 kg. Compared with the starting point, there was a significant decrease of −0.8 ± 1.3 kg, and a significant decrease compared with the control group (*p* = 0.020). In the tomato group, the starting point, percentage of fat was 35.0 ± 2.9 at week 0, 35.1 ± 2.8 at 4 weeks, and 34.7 ± 2.6 at week 8. Compared with the baseline, there was a significant drop in the percentage of fat to −0.3% ± 0.8%, and there was a significant decrease compared with the control group (*p* = 0.008) at week 8. In the control group, the starting point of the waist circumference was 90.9 ± 5.5 cm, then 90.1 ± 9.7 cm at week 4, and 89.2 ± 9.0 cm at week 8. In the tomato group, corresponding values of the waist circumference were 89.0 ± 8.2, 87.4 ± 5.3, and 86.2 ± 5.2 cm. Compared with the starting point, there was a significant decrease of −2.9 ± 1.7 cm, and a significant decrease compared with the control group (*p* = 0.018). In the tomato group, the starting point, percentage of hip circumference was 101.0 ± 3.8 at week 0, 99.8 ± 3.6 at 4 weeks, and 99.3 ± 3.6 at week 8. Compared with the baseline, there was a significant drop in the percentage of hip circumference to −1.7% ± 1.1%, and there was a significant decrease compared with the control group (*p* = 0.010).

As shown in Table 6, after 8 weeks of the dietary intervention, the tomato group had significantly lower serum total cholesterol, triglyceride, HDL-C, and systolic blood pressure but not in the control group. The tomato group showed significant declines in TC of −6.1 ± 36.8 mg/dL (*p* = 0.041), TG of −25.9 ± 49.4 mg/dL (*p* = 0.023), and SBP of −5.5 ± 14.2 mmHg (*p* = 0.042), and an increase in HDL-C of 0.83 ± 14.9 mg/dL (*p* = 0.019) over the control group. Both groups had significant decreases in the inflammatory marker, IL-6, but there was no significant difference between the two groups (*p* = 0.544).

## 4. Discussion

The *Lycopersicon esculentum* Mill. tomato is a unique species in Taiwan that has a greenish-red skin, a sweet and sour flavor, and moderate hardness, and can be eaten raw. The tomato is low-calorie, fiber-rich, and juicy, and it can reduce daily caloric intake and increase a subject’s satiety. After 8 weeks of the experiment, the tomato group had significantly increased dietary vitamin A intake compared with the control group. Blood biomarkers of FRAP, β-carotene, and lycopene had significantly improved (Figure 1), confirming that subjects had good compliance. Although the bioavailability of lycopene may be higher after tomatoes are cooked than raw, eating raw tomatoes can maintain nutrients, and the lipid content of meals can also help lycopene absorption. The intervention was not a major adjustment of daily dietary habits or nutrient intake. After the intervention period, the calorie intake of the two groups decreased significantly. A possible reason is that the tomato diet and the control diet were low-calorie-dense diets that replaced high-calorie diets, thereby reducing daily calorie intake. However, there were no significant differences in crude fiber or dietary fiber between the two groups after 8 weeks.

Tomatoes contain many natural antioxidants that are beneficial to health. The most well-known one is lycopene, whose molecular structure contains a number of conjugated double bonds [13]. Due to its structural characteristics, it can be used as a free radical scavenger to inhibit oxidative damage caused by free radicals [14]. The antioxidant capacity of FRAP is twice that of β-carotene and 10 times that of α-tocopherol [15]. According to our results, tomato intake led to significantly increased FRAP (*p* < 0.05, Figure 1), which was probably related to the increase in serum β-carotene and lycopene concentration in tomato intake.

The experimental results show that both the control and tomato groups significantly decreased the inflammatory marker, IL-6 (Table 6). IL-6 stimulates acute inflammatory response proteins and inflammatory cytokines, and high-sensitivity C-reactive protein concentrations significantly improved [16,17,18]. Epidemiological investigations have determined that the consumption of a variety of vegetables and fruits can reduce body weight and body fat mass [19]. In this study, overweight postmenopausal women were given a daily intake of about 4 to 5 servings of vegetables and two servings of fruit, and the body weight and fat mass of the tomato group decreased significantly after 8 weeks. The reason was that the daily caloric intake, fat, and carbohydrates significantly decreased by guiding the correct nutritional concept and combining diet intervention. Dietary fiber in the diet has the effect of reducing body fat; a previous study showed that the intake of a low-fat diet by postmenopausal women improved daily dietary fiber by 7 g and reduced body fat by 2.7% [20]. However, there was no statistical difference in dietary fiber between the tomato group and the control group in our study. As for the experimental results of subjects, the waist circumference and waist–hip ratio of subjects also exhibited the phenomenon of fat accumulation in the abdomen. Reducing abdominal fat can reduce the risks of CVD. In a previous study, 42 obese adults consumed 8 servings daily of vegetables and 2~3 servings of fruit for one year; there was a significant increase in serum carotenoids and decreases in weight and body fat [21].

After 8 weeks, the tomato diet group showed effective reductions in blood glucose (Table 4). This is due to a reduction in the meal glycemic load, which lowers blood glucose [22]. Greater dietary fiber intake can delay gastric emptying, and lower GI values can reduce blood glucose after a meal [2,4]. However, the tomato diet group had no significant changes in insulin concentration, resistance, or sensitivity, indicating that there was no change in insulin resistance or sensitivity in the tomato diet group. The results of this experiment are consistent with previous studies [23,24].

Lycopene from tomatoes can reduce serum cholesterol by inhibiting HMG-CoA reductase, adjusting the LDL receptor, and inhibiting acyl-CoA cholesterol acyltransferase activity [25]. On the other hand, lycopene can reduce serum cholesterol by activating the ATP-binding cassette transporter 1 (ABCA1) in the liver and increasing cholesterol excretion [26,27]. In addition, an increase in the dietary intake of soluble fiber can increase the binding of fiber and cholesterol in the small intestine to increase the cholesterol efflux effect [26]. In a randomized single-blind controlled trial, 250 mL of tomato juice (containing 41.8 mg lycopene) reduced serum TC by 3% after 2 weeks by inhibiting cholesterol biosynthesis and increasing cholesterol excretion to lower blood cholesterol concentrations [27].

TG levels were significantly decreased in the tomato intervention group after 8 weeks of supplementation (Table 6). This can be explained as lycopene can improve liver lipid metabolism, inhibit lipid peroxidation, and then reduce blood TG levels. The metabolites of lycopene can be activated in the liver, and PPAR-α can increase β-oxidation, which may lead to lower TG levels in the liver and blood. 9-Oxo-10 (E), 12 (E)-octadecadienoic acid from tomatoes is an agonist of PPAR-α that could improve abnormal fat metabolism to achieve an antidyslipidemia effect [28]. In clinical research, six healthy adults were provided tomato soup and canned tomatoes every day, which contained 46 mg of lycopene. After seven days, the TG levels of these adults significantly decreased [15].

In the current study, after 8 weeks of dietary intervention, there was no significant change in LDL-C in the tomato group. One study confirmed that an increase in tomato intake can effectively reduce the risk of coronary disease [29], whereas previous studies showed that the consumption of tomato lycopene can increase the concentration of lycopene in the body and prevent LDL oxidation [26]. In vivo lipid peroxidation of metabolites are cytotoxic and genotoxic and play important roles in degenerative diseases [30]. Oxidized LDL-C forms foam cells during arterial plaque formation, recognized as a risk factor for coronary heart disease [31]. It is recommended to take 40 mg of lycopene and various tomato products daily, which can effectively reduce LDL-C [30]. In this study, only about 11 mg of lycopene was taken every day. It is speculated that the insufficient dose of lycopene in this experiment has no effect on lowering LDL-C [32].

High blood pressure is caused by menopause and estrogen hormones, and blood pressure in premenopausal women is lower than that in men, but the prevalence of hypertension in postmenopausal women is greater than that in men, especially SBP when compared with males of the same age [33]. The American Heart Association pointed out that 75% of U.S. postmenopausal women have hypertension [3]. Taiwan’s Department of Health statistics shows that only 2.4% of 31~44-year-old women have hypertension, which increases with age, and the prevalence in women over 65 years soars to 52.3% [34]. In the experimental results, the SBP of the tomato group significantly decreased. Hypertension can be corrected and improved by healthy eating behaviors; the treatment of hypertension in the Dietary Approaches to Stop Hypertension (DASH) recommends a high intake of fruits and vegetables. This study showed that SBP decreased by 2.8 mmHg, and DBP decreased by 1.1 mmHg after 8 weeks. The subjects increased their intake of fruits and vegetables, which increased their intake of dietary potassium and reduced their sodium intake, but only SBP was significantly reduced, and DBP remained unchanged. Presumably, the reason is that their intake of fruits and vegetables was less than the DASH diet.

## 5. Conclusions

In conclusion, fresh tomato intervention for 8 weeks significantly lowered serum total cholesterol, triglyceride, systolic blood pressure and blood sugar, and higher high-density lipoprotein cholesterol and antioxidant biomarkers. Therefore, fresh tomato intervention could improve metabolic syndromes in overweight postmenopausal women.

## Figures and Tables

**Figure 1 biology-13-00588-f001:**
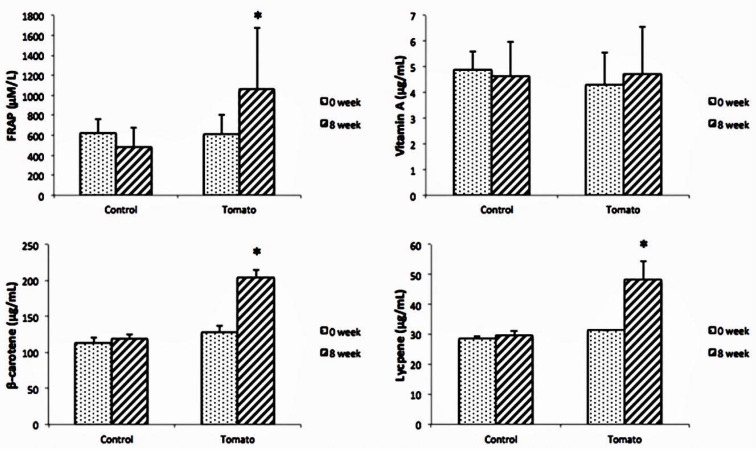
Effect of tomato on the ferric-reducing ability of plasma (FRAP), and serum vitamin and carotenoid concentrations during intervention period. Data are mean ± SD; *n* = 23 and 30 for the control and tomato groups, respectively. * significant difference from baseline; *p*-value < 0.05.

**Table 1 biology-13-00588-t001:** The nutrients of control and tomato (per 100 g).

	Control	Tomato
Energy (kcal)	31.0	23.1
Protein (g)	3.1	0.9
Carbohydrate (g)	4.8	4.9
Fat (g)	0.5	0.2
Fiber (g)	0.6	0.6
Dietary fiber (g)	1.7	1.2
Vitamin A (RE)	0.0	84.2
Vitamin C (mg)	183.6	21
Lycopene (mg)	0.0	5.5

**Table 2 biology-13-00588-t002:** Baseline characteristics of subjects.

	Control, *n* = 23	Tomato, *n* = 30	*p* ^2^
Age (y)	59.2 ± 8.0	60.2 ± 5.9	0.637
Height (cm)	156.3 ± 5.7	155.6 ± 4.1	0.628
Weight (kg)	66.1 ± 8.4	63.9 ± 5.8	0.334
BMI (kg/m^2^)	27.0 ± 2.2	26.4 ± 2.1	0.375
WC (cm)	90.9 ± 5.5	89.0 ± 8.2	0.243
HC (cm)	120.4 ± 5.1	101.0 ± 3.8	0.315
WHR	0.9 ± 0.0	0.9 ± 0.1	0.603
SBP (mmHg)	140.5 ± 20.6	132.8 ± 18.0	0.195
DBP (mmHg)	79.8 ± 10.4	78.8 ± 8.3	0.722
FSH (mIU/mL)	57.9 ± 29.8	62.9 ± 23.3	0.129
E2 (pg/mL)	14.9 ± 8.1	11.4 ± 6.5	0.235

**Table 3 biology-13-00588-t003:** Daily nutrient intakes of control and tomato groups.

	Control, *n* = 23	Tomato, *n* = 30	*p* ^2^
Energy (kcal)			
0 week	1639.0 ± 61.7	1607.3 ± 72.7	0.173
4 weeks	1606.9 ± 68.9 *	1557.3 ± 66.4 *	0.033
8 weeks	1571.0 ± 64.9 *	1522.2 ± 60.5 *	0.024
Protein (g)			
0 week	60.9 ± 8.5	54.3 ± 11.2	0.061
4 weeks	62.3 ± 10.8	56.1 ± 15.5	0.186
8 weeks	65.8 ± 12.5	55.8 ± 12.7	0.021
Fat (g)			
0 week	59.9 ± 11.1	57.1 ± 8.2	0.382
4 weeks	59.6 ± 10.7	54.5 ± 7.2	0.085
8 weeks	55.3 ± 10.4	53.0 ± 7.4 *	0.43
Carbohydrate (g)			
0 week	219.7 ± 25.3	223.7 ± 31.9	0.688
4 weeks	211.0 ± 37.0	215.3 ± 30.4	0.697
8 weeks	207.8 ± 30.5	210.1 ± 24.8 *	0.8
Fiber (g)			
0 week	5.9 ± 2.5	5.9 ± 3.5	0.997
4 weeks	5.9 ± 1.7	6.6 ± 2.5	0.332
8 weeks	6.1 ± 1.8	6.7 ± 2.7	0.42
Dietary fiber (g)			
0 week	19.8 ± 7.6	18.8 ± 6.4	0.664
4 weeks	22.0 ± 12.3	21.4 ± 6.8	0.859
8 weeks	20.2 ± 5.4	20.5 ± 6.4	0.879
Vitamin A (RE)			
0 week	837.5 ± 712.0	1553.0 ± 2170.8	0.056
4 weeks	684.8 ± 614.3	2005.6 ± 2782.1	0.02
8 weeks	1014.9 ± 1204.9	2174.9 ± 2907.8	0.039
Vitamin E (alpha-TE)			
0 week	5.4 ± 1.5	5.0 ± 1.1	0.932
4 weeks	5.8 ± 1.7	5.4 ± 1.2	0.668
8 weeks	5.6 ± 1.6	5.6 ± 1.5	0.157
Vitamin C (mg)			
0 week	207.5 ± 192.6	198.5 ± 146.8	<0.001
4 weeks	535.1 ± 104.9	226.4 ± 109.9	<0.001
8 weeks	538.6 ± 113.5	258.1 ± 142.5	<0.001

All values are mean ± SD. * Differences between groups were assessed using independent *t*-test (*p* < 0.05 indicates statistically significant).

**Table 4 biology-13-00588-t004:** Effect of tomato on glycemic control profile.

	Control, *n* = 23	Tomato, *n* = 30	*p* ^2^
Blood glucose (mg/dL)			
0 week	101.7 ± 33.1	103.1 ± 13.2	0.645
8 weeks	103.0 ± 24.0	98.5 ± 13.3 *	0.975
Δ 0–8 weeks	1.3 ± 10.6	−5.2 ± 13.6	0.043
Insulin (mU/dL)			
0 week	10.7 ± 5.6	9.3 ± 4.2	0.092
8 weeks	11.5 ± 4.7	8.4 ± 3.7	0.107
Δ 0–8 weeks	0.8 ± 5.6	0.6 ± 3.3	0.885
HOMA-IR			
0 week	2.9 ± 2.1	2.1 ± 1.3	0.138
8 weeks	3.0 ± 1.7	2.4 ± 1.3	0.185
QUICKI			
0 week	0.3 ± 0.0	0.3 ± 0.0	0.280
8 weeks	0.3 ± 0.0	0.4 ± 0.0	0.225

All values are mean ± SD. Differences between groups were assessed using independent *t*-test (*p* < 0.05 indicates statistically significant). * Differences in plasma glucose change from baseline within the dietary intervention group were assessed by paired *t*-test. *p* < 0.05 indicates statistically significant. HOMA-IR, homeostasis model assessments–insulin resistance; QUICKI, quantitative insulin sensitivity check index.

**Table 5 biology-13-00588-t005:** Effect of tomato on the changes in anthropometric measurements during intervention period1.

	Control, *n* = 23	Tomato, *n* = 30	*p* ^2^
Weight (kg)			
4–0 week	−0.5 ± 0.7	−0.7 ± 0.7	0.279
8–0 week	−0.7 ± 0.9	−1.1 ± 1.0	0.284
BMI (kg/m^2^)			
4–0 week	−0.2 ± 0.3	−0.3 ± 0.3	0.085
8–0 week	−0.3 ± 0.4	−0.4 ± 0.4	0.223
Fat mass (kg)			
4–0 week	−0.2 ± 1.0	−0.4 ± 1.2	0.498
8–0 week	−0.4 ± 1.1	−0.8 ± 1.3	0.020
Fat (%)			
4–0 week	−0.2 ± 1.4	0.1 ± 0.6	0.423
8–0 week	−0.2 ± 1.5	−0.3 ± 0.8	0.008
WC (cm)			
4–0 week	−1.3 ± 1.3	−1.7 ± 1.4	0.727
8–0 week	−2.2 ± 2.5	−2.9 ± 1.7	0.018
HC (cm)			
4–0 week	−0.3 ± 1.6	−1.3 ± 1.0	0.001
8–0 week	−1.5 ± 1.8	−1.7 ± 1.1	0.010
WHR			
4–0 week	0.0 ± 0.2	0.0 ± 0.2	0.333
8–0 week	0.0 ± 0.0	0.0 ± 0.0	0.177

All values are mean ± SD. Differences between groups were assessed using independent *t*-test (*p* < 0.05 indicates statistically significant). BMI, body mass index; WC, waist circumference; HC, hip circumference; WHR, waist–hip ratio.

**Table 6 biology-13-00588-t006:** Effect of tomato on serum lipids and blood pressure during intervention period.

	Control, *n* = 23	Tomato, *n* = 30	*p* ^2^
TC (mg/dL)			
0 week	201.0 ± 36.8	216.4 ± 32.9	0.146
8 weeks	197.9 ± 38.3	210.3 * ± 34.7	0.368
Δ 0–8 weeks	−2.9 ± 19.2	−6.1 ± 36.8	0.041
TG (mg/dL)			
0 week	109.4 ± 62.7	135.8 ± 50.1	0.128
8 weeks	114.6 ± 75.1	113.6 * ± 49.4	0.293
Δ 0–8 weeks	−4.5 ± 41.1	−25.87 ± 49.4	0.023
LDL-C (mg/dL)			
0 week	136.1 ± 34.3	140 ± 31.6	0.573
8 weeks	130.6 ± 34.4	141.8 ± 27.1	0.681
Δ 0–8 weeks	−6.2 ± 18.4	−2.7 ± 27.4	0.546
HDL-C (mg/dL)			
0 week	59.0 ± 14.7	63.2 ± 14.0	0.223
8 weeks	57.6 ± 13.1	64.4 ± 13.6	0.311
Δ 0–8 weeks	−2.5 ± 14.1	0.83 ± 14.9	0.019
SBP (mmHg)			
0 week	135.3 ± 21.5	133.4 ± 17.7	0.195
8 weeks	135.6 ± 18.8	128.9 * ± 13.1	0.103
Δ 0–8 weeks	−0.3 ± 19.8	−5.5 ± 14.2	0.042
DBP (mmHg)			
0 week	77.3 ± 10.5	78.6 ± 8.1	0.722
8 weeks	77.7 ± 10.3	77.2 ± 6.8	0.283
Δ 0–8 weeks	0.4 ± 10.4	−1.2 ± 7.2	0.764
IL-6 (pg/mL)			
0 week	431.3 ± 322.6	433.85 ± 297.3	0.544
8 weeks	61.7 * ± 72.6	11.92 * ± 29.4	0.088
Δ 0–8 weeks	−429.2 ± 247.0	−421.9 ± 284	0.449

All values are mean ± SD. Differences between groups were assessed using independent t-test (*p* < 0.05 indicates statistically significant). * Differences in intake change from baseline within the dietary intervention group were assessed by paired t-test. *p* < 0.05 indicates statistically significant. TC, total cholesterol; TG, triglyceride; LDL-C, low-density lipoprotein cholesterol; HDL-C, high-density lipoprotein cholesterol; SBP, systolic blood pressure; DBP, diastolic blood pressure.

## Data Availability

Date is contained within the article.

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
