# Peer review of "Fresh Tomato (Lycopersicon esculentum Mill.) in the Diet Improves the Features of the Metabolic Syndrome: A Randomized Study in Postmenopausal Women"

_biology, 2024, doi:10.3390/biology13080588_

Round 1

Reviewer 1 Report

Comments and Suggestions for Authors

Strengths and Limitations of study

Strength: The manuscript aims to find the effect of raw tomato eating on metabolic syndrome. Managing metabolic syndrome is crucial because it significantly increases the risk of cardiovascular diseases and type 2 diabetes, which are major health concerns globally. Improving metabolic syndrome symptoms through dietary intervention is certainly a strength.

Limitation: The study addresses an important topic in the field of dietary intervention with tomatoes. The methodology is robust and well-executed. Although the manuscript discusses valuable insights into the effects of intervention over a short period; however, the lack of long-term follow-up data limits the understanding of sustained impacts and potential long-term effects.

Major comments:

1.      Several studies have reported that tomatoes aid in weight loss and possess anti-inflammatory properties. Please explain the novelty of your study.

2.      Numerous studies have utilized tomatoes in various diseases. Why did you choose raw tomatoes instead of puree or other forms?

3.      Dietary interventions are often more effective when combined with physical activity. Did your study include any guidelines regarding physical activity?

4.      IL-6 is a multifunctional cytokine. Please explain your rationale for selecting IL-6 over other anti-inflammtory cytokines.

5.      The study reports the levels of FSH and E2 at the beginning. Please explain why you did not measure the levels of FSH and estradiol at weeks 4 and 8.

6.      According to the data availability statement (line no. 312), no new data were created or analyzed in this study, yet you reported conducting research. Please clarify this discrepancy.

7.      Consuming raw tomatoes can cause side effects, including digestive issues, urinary problems, and kidney stones. Did you monitor these parameters in the study subjects?

Minor comments

1.      Make it control and tomato diet in Line no. 124.

2.      Arrange reference according to journal format.

3.      Improvise title of Table 3.

4.      Describe week no. in line no. 187.

5.      Provide the catalog no. of commercial kit in line no. 112.

6.      Please elaborate methodology used for ELISA in section 2.3.

7.      Section 3.3 is labeled twice, please make corrections accordingly.

Comments on the Quality of English Language

While the manuscript is generally well-written, thorough proofreading for minor grammatical errors and typos is recommended.

Author Response

Major comments:

Comment 1.      Several studies have reported that tomatoes aid in weight loss and possess anti-inflammatory properties. Please explain the novelty of your study.

Response 1: Black persimmon (Lycopersicon esculentum Mill) is the most traditional tomato variety in Taiwan. The pulp of black persimmon tomatoes is relatively hard and has high acidity, so it is very suitable for stir-fry, soup base, or tomato scrambled eggs. Its sour taste can add a special flavor to dishes. In Kaohsiung and Tainan in southern Taiwan, black persimmons (Lycopersicon esculentum Mill) are also cut into plates and mixed with minced ginger and soy sauce to become a famous snack called "tomato cut plate." The pulp of black persimmon tomatoes is relatively hard and has high acidity, so it is very suitable for stir-fry, soup base or tomato scrambled eggs. Its sour taste can add a special flavor to dishes.

The novelty of this study was the beneficial effects of fresh tomato (Lycopersicon esculentum Mill, a native Taiwan variety) consumption on metabolic syndrome in overweight women after menopause remains unclear.

Comment 2.      Numerous studies have utilized tomatoes in various diseases. Why did you choose raw tomatoes instead of puree or other forms?

Response 2: We revise the introduction section Line 67.

Comment 3.      Dietary interventions are often more effective when combined with physical activity. Did your study include any guidelines regarding physical activity?

Response 3: Our study has included the physical activity guidelines. Questionnaires were used to assess their smoking, drinking and exercise habits. Smoking and drinking, including frequency, amount of food consumed each time, and type. Exercise habits will record the exercise items and time. Maintain exercise regularly before and after the experiment. We revise on 2.3. Data Collection.

Comment 4.      IL-6 is a multifunctional cytokine. Please explain your rationale for selecting IL-6 over other anti-inflammatory cytokines.

Response 4: IL-6 can have anti-inflammatory effects. Classic IL-6 signaling via transmembrane IL-6 receptors (IL-6R) drives processes like regeneration of intestinal epithelial cells (which is crucial for gut health), activation of the hepatic acute phase response, and inhibition of epithelial cell apoptosis1. So, it’s not just about inflammation—it’s also about healing and tissue repair.

Comment 5.      The study reports the levels of FSH and E2 at the beginning. Please explain why you did not measure the levels of FSH and estradiol at weeks 4 and 8.

Response 5: The levels of FSH and E2 at the beginning were confirmed to meet the criteria for postmenopausal women.

Comment 6.      According to the data availability statement (line no. 312), no new data were created or analyzed in this study, yet you reported conducting research. Please clarify this discrepancy.

Response 6: We confirm this.

Comment 7.      Consuming raw tomatoes can cause side effects, including digestive issues, urinary problems, and kidney stones. Did you monitor these parameters in the study subjects?

Response 7: These side effects are monitored on case report forms.

Minor comments

Comment 1.      Make it control and tomato diet in Line no. 124.

Response 1: We revised it.

Comment 2.      Arrange references according to journal format.

Response 2: We revised it.

Comment 3.      Improvise the title of Table 3.

Response 3: We revised it.

Comment 4.      Describe week no. in line no. 187.

Response 4: We revised it.

Comment 5.      Provide the catalog no. of the commercial kit in line no. 112.

Response 5: We revised it.

Comment 6.      Please elaborate on the methodology used for ELISA in section 2.3.

Response 6: We revised it in section 2.3.

Comment 7.      Section 3.3 is labeled twice; please make corrections accordingly.

 Response 7: We revise it. Correct 3.4. Oxidative-Stress-Related Markers

Reviewer 2 Report

Comments and Suggestions for Authors

No comments or suggestions for the authors.

The paper is good

Author Response

Response: Thank you for your comments.

Reviewer 3 Report

Comments and Suggestions for Authors

I have read the paper entitled “Fresh Tomato (Lycopersicon esculentum Mill) in the Diet Improves the Features of the Metabolic Syndrome: A Randomized Study in Postmenopausal Women” by Chen and Chien.

The paper investigates the impact of tomato consumption on reducing metabolic syndrome risk factors in overweight postmenopausal women.

The paper is well constructed, but this reviewer has some concerns:

1)      The BMI of postmenopausal women is just above the normal range, which is between 18.50 and 24.99. Therefore, I would not refer to them as overweight but would nuance the concept.

2)       Furthermore, statistical significance is not always correctly represented and is difficult to interpret. For example, among other issues: in Table 4, blood glucose at 8 weeks is marked with an asterisk as if it were significant, but the p-value is 0.975, whereas the difference in the same parameter between 0 and 8 weeks is not marked with an asterisk but has a p-value of 0.043. Additionally, on page 7 of the text, regarding Figure 1, the authors write: 'At week 8, FRAP was significantly decreased to 485.6 ± 192.4 μM/L in the control group (p = 0.012) and increased to 1066.4 ± 609.6 μM/L in the tomato group (p = 0.002).'  However, the decrease in FRAP in the control group is not highlighted as statistically significant in the graph as in the tomato group.

Authors have to carefully review the tables and figures to ensure they are easily understandable and, most importantly, aligned with the text.

Author Response

Comment 1)      The BMI of postmenopausal women is just above the normal range, between 18.50 and 24.99. Therefore, I would not refer to them as overweight but would nuance the concept.

Response 1): The baseline characteristics of the study participants' shown in Table 2. All participants (53 postmenopausal women with metabolic syndrome) completed the study. According to the definition of overweight by the Ministry of Health and Welfare in Taiwan, overweight (24.0 ≦ BMI < 27.0) and obese (BMI ≧ 27.0). The average BMI was 27.0 ± 2.2 in the control group and 26.4 ± 2.1 in the tomato group.

Comment 2)       Furthermore, statistical significance is not always correctly represented and is difficult to interpret. For example, among other issues, in Table 4, blood glucose at 8 weeks is marked with an asterisk as if it were significant, but the p-value is 0.975. In contrast, the difference in the same parameter between 0 and 8 weeks is not marked with an asterisk but has a p-value of 0.043. Additionally, on page 7 of the text, regarding Figure 1, the authors write: 'At week 8, FRAP was significantly decreased to 485.6 ± 192.4 μM/L in the control group (p = 0.012) and increased to 1066.4 ± 609.6 μM/L in the tomato group (p = 0.002).'  However, the decrease in FRAP in the control group is not highlighted as statistically significant in the graph as in the tomato group.

Authors must carefully review the tables and figures to ensure they are easily understandable and, most importantly, aligned with the text.

Response 2): We checked all tables and Figure 1.* Differences in intake change from baseline within the dietary intervention group were assessed by paired t-test in Table 4. At week 8, there was no significant difference between the two groups at 103.0 ± 24.0 and 98.5 ± 13.3 mg/dL (p = 0.975), but the tomato group had significantly decreased compared to week 0 (p = 0.0 22).

Figure 1. At week 8, there was no significant difference between the two groups at 103.0 ± 24.0 and 98.5 ± 13.3 mg/dL (p = 0.975), but the tomato group had significantly decreased compared to week 0 (p = 0.0 22).